# ATP binds and inhibits the neurodegeneration-associated fibrillization of the FUS RRM domain

Jian Kang[1], Liangzhong Lim[1] & Jianxing Song[1]

Adenosine triphosphate (ATP) provides energy for cellular processes but has recently been found to act also as a hydrotrope to maintain protein homeostasis. ATP bivalently binds the disordered domain of FUS containing the RG/RGG sequence motif and thereby affects FUS liquid-liquid phase separation. Here, using NMR spectroscopy and molecular docking studies, we report that ATP specifically binds also to the well-folded RRM domain of FUS at physiologically relevant concentrations and with the binding interface overlapping with that of its physiological ssDNA ligand. Importantly, although ATP has little effect on the thermodynamic stability of the RRM domain or its binding to ssDNA, ATP kinetically inhibits the RRM fibrillization that is critical for the gain of cytotoxicity associated with ALS and FTD. Our study provides a previously unappreciated mechanism for ATP to inhibit fibrillization by specific binding, and suggests that ATP may bind additional proteins other than the classic ATP-dependent enzymes.

---

[1] Department of Biological Sciences, Faculty of Science, National University of Singapore, 10 Kent Ridge Crescent, Singapore 119260, Singapore. Correspondence and requests for materials should be addressed to J.S. (email: dbssjx@nus.edu.sg)

Adenosine triphosphate (ATP) is a unique nucleotide acting as the "molecular currency" of cellular energy transfer that provides energy to drive a myriad of physiological processes in all living cells, including muscle contraction, nerve impulse propagation, and chemical synthesis[1–3]. Intriguingly, although ATP-dependent proteins/enzymes require only micromolar concentrations of ATP, the cellular concentrations of ATP are very high, which range from 1 to 12 mM depending on cell types[1–3]. So a fundamental question is why cells invest so much energy to maintain such high ATP concentrations. Previously, it was proposed that the free-energy difference between ATP and ADP is required to drive ATP-dependent reactions and consequently a ~50-fold higher ATP/ADP ratio is necessary to fuel a variety of metabolic reactions. Consequently, cells need to maintain ATP at the millimolar range but ADP and AMP at <50 µM and <1–10 µM, respectively[1–3]. Nevertheless, it is of both fundamental and physiological significance to understand whether at such high concentrations ATP can have physiological roles other than acting as the "universal energy currency."

Only recently, it has been determined that at high concentrations ~>6 mM, ATP acts as a biological hydrotrope to dissolve liquid–liquid phase separation (LLPS) of RNA-binding proteins (RBPs) including FUS, as well as to generally prevent/dissolve protein aggregates and amyloid fibrils[2,3]. These new roles of ATP were proposed to result from the hydrotropic properties of ATP, which is composed of two major components: the relatively hydrophobic aromatic ring of adenine and highly polar triphosphate chain. The aromatic ring is capable of becoming clustered over hydrophobic patches of liquid droplets or aggregates, while the triphosphate chain strongly interacts with water molecules, thus leading to dissolution of LLPS or protein aggregates[2,3]. Therefore, it is now established that, in addition to servicing as the universal energy source for various biological reactions, for which micro-molar concentrations are sufficient, ATP also plays a central role in promoting protein solubility that needs millimolar concentrations[2,3]. Very recently, we found that, at concentrations 1–4 mM, ATP could in fact induce LLPS of the FUS C-terminal RG/RGG-rich domain, which is intrinsically disordered and unable to phase separate by itself. We thus proposed that in addition to act as a hydrotropic molecule, ATP can also behave as a bivalent binder to modulate LLPS[4]. Nevertheless, so far, no study has addressed whether ATP can specifically interact with the well-folded proteins other than the classic ATP-dependent proteins/enzymes.

In the current study, we aimed to address this question by characterizing the interaction between ATP and the well-folded RNA-recognition motif (RRM) domain of FUS. FUS is a 526 residue protein intrinsically prone to aggregation[5–8], which is composed of an N-terminal low-sequence complexity (LC) domain (1–267); an RRM domain (285–370) capable of binding a large array of RNA and DNA; and C-terminal LC domain that contains many RG/RGG sequence motifs (371–526). Very interestingly, the intrinsically disordered RG/RGG regions of FUS were recently found to bind various nucleic acids including DNA and RNA with the degenerative sequence specificity[9,10]. Furthermore, FUS aggregation has been observed in amyotrophic lateral sclerosis (ALS), frontotemporal dementia (FTD), and the polyglutamine diseases, which include Huntington disease, spinocerebellar ataxia, and dentatorubropallidoluysian atrophy[5–8]. These results suggest that FUS aggregation might have a general role in various neurodegenerative diseases.

The RRM domain is one of the most abundant domains in eukaryotes, and most heterogeneous nuclear ribonucleoproteins (hnRNPs) contain one or several RRM domains that mediate the direct interaction with various nucleic acids to control both RNA processing and gene expression[11–14]. Previously the structure of the FUS RRM domain has been determined by nuclear magnetic resonance (NMR) spectroscopy to adopt the same overall fold as other RRMs[13], which consists of a four-stranded β-sheet and two perpendicular α-helices. Previous in vivo studies also revealed that, different from TDP-43, only the prion-like domain (PLD) of FUS is not sufficient for manifesting cytotoxicity and its RRM domain is also required[8]. Recently, we showed that the FUS RRM domain has relatively low thermodynamic and dynamic stabilities and could spontaneously self-assemble into amyloid fibrils[14].

In the present study, we aimed to characterize the binding interaction between FUS RRM and ATP by NMR spectroscopy and molecular docking. NMR spectroscopy is very powerful in not only detecting but also quantifying residue-specific parameters for the very weak binding events including the formation of "fuzzy complex" and LLPS of FUS[15–18], for which other biophysical methods such as isothermal titration calorimetry are no longer suitable. Our study revealed, for the first time to our knowledge, that ATP could specifically bind to a pocket of the FUS RRM domain with the average dissociation constant (Kd) of $3.77 \pm 0.49$ (mM). Further studies with AMP and triphosphate acid (PPP) showed that both adenine and triphosphate chain are needed for this binding: the aromatic ring of adenine sits in a relatively hydrophobic pocket with the direct contact with Arg328 side chain likely by π–cation interaction, while triphosphate chain is located in a positively charged pocket by electrostatic interaction. Interestingly, although ATP has no detectable effect on the binding of the FUS RRM domain with its physiological ligand, the human telomeric single-stranded DNA (TssDNA), as well as on its thermodynamic stability, ATP appears to kinetically inhibit the self-assembly of the FUS RRM domain into amyloid fibrils. Therefore, our study not only provides a novel mechanism for ATP to specifically bind and kinetically inhibit the formation of amyloid fibril of the FUS RRM domain associated with its cytotoxicity but also implies that ATP may bind a large array of previously unknown well-folded proteins other than the ATP-dependent proteins/enzymes to regulate cell physiology.

## Results

**ATP binds RRM at physiologically relevant concentrations.** Here, to assess whether ATP (Fig. 1a) interacts with the FUS RRM domain (Fig. 1b), we acquired two-dimensional $^1$H-$^{15}$N NMR heteronuclear single quantum coherence (HSQC) spectra of the $^{15}$N-labeled FUS RRM domain at 40 µM in 10 mM sodium phosphate buffer containing 150 mM NaCl (pH 6.8), titrated with ATP at 14 concentrations ranging from 0 to 40 mM. By superimposing HSQC spectra in the presence of ATP at different concentrations, we found that only a small set of HSQC peaks showed a meaningful shift upon gradual addition of ATP with the concentration up to 40 mM (Fig. 2a and Supplementary Fig. 1). With our previous sequential assignment[14], we successfully mapped out the chemical shift difference (CSD) of the residues between the free state and those in the presence of ATP at different concentrations (Fig. 2b). Interestingly, only 10 residues located over five regions of the FUS RRM sequence have large shifts, with CSD values >0.065 (the average value + one standard deviation). These residues include Asn284, Asn285, Val289, Thr313, Asn314, Thr326, Arg328, Thr338, Thr370, and Arg371. As judged by the CSD traces of the 10 residues (Fig. 2c), the binding of ATP to FUS RRM is largely saturated at 40 mM, thus suggesting that ATP does specifically interact with the FUS RRM domain[15–17]. As a consequence, we successfully obtained the residue-specific dissociation constant (Kd) values of 10 residues by fitting the CSD traces to the one-site binding model[15–17]. Strikingly, the Kd values range from 3.04 mM of Asn284 to

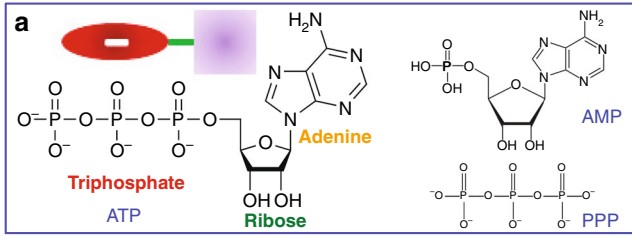

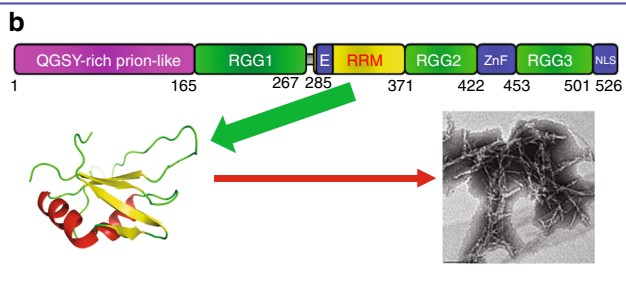

**Fig. 1** Chemical structure of adenosine triphosphate (ATP) and three-dimensional structure of FUS RNA-recognition motif (RRM). **a** Chemical structure of ATP, AMP, and triphosphate acid (PPP). Inlet: a cartoon model of ATP showing its triphosphate chain, which is highly polar and negatively charged, and relatively hydrophobic aromatic ring of adenine base linked by a ribose. **b** 526residue FUS is composed of: N-terminal low-sequence complexity (LC) region (1–267) including a QGSY-rich prion-like domain (PLD), and an RG/RGG-rich region (RGG1); RRM (285–370); and C-terminal domain CTD (371–526) that contains an RG/RGG-rich region (RGG2), a zinc finger (ZnF), and another RG/RGG-rich region (RGG3) carrying a nuclear localization signal (NLS). RRM domain adopts a typical RRM fold consisting of a four-stranded β-sheet and two perpendicular α-helices, which can spontaneously self-assemble into amyloid fibrils

4.43 mM of Thr370 (Fig. 2c), with the average Kd value for the 10 residues of 3.77 ± 0.49 mM. This result thus indicates that ATP is capable of specifically binding to the FUS RRM domain at physiologically relevant concentrations of ATP that range from 1 to 12 mM[1–3].

**Structural requirement for ATP to specifically bind RRM.** Owing to the extremely low binding affinity, it is impossible to determine the three-dimensional structure of the ATP-RRM complex by classic NMR spectroscopy or X-ray crystallography. Therefore, to visualize its structure, we conducted molecular docking by the well-established HADDOCK program[19] with the NMR constraints derived from HSQC titrations as we previously performed on the EphA4 small molecule complex[20]. Figure 3 presents the lowest energy docking model of the ATP-RRM complex. As seen in Fig. 3a, b, ATP binds to a pocket constituted by the 10 residues with highly shifted HSQC peaks (CSD > 0.065). All ten residues are located over the N- and C-termini and loops, except for Val289 located within the first β-strand and Thr338 within the third β-strand. Interestingly, the oxygen atoms of triphosphate chain of ATP establish two hydrogen bonds with the side chain protons of Asn323 and Thr338, respectively (Fig. 3c). Further examination revealed that the aromatic ring of ATP is located in a pocket of the FUS RRM domain with the surface relatively hydrophobic (Fig. 3d, e), and interestingly, the aromatic ring of ATP also has direct contact with the side chain of Arg328, likely by the π–cation interaction[21,22]. On the other hand, the triphosphate chain of ATP appears to be embedded in a pocket of the FUS RRM domain with the negatively charged surface (Fig. 3d, e) by electrostatic interaction.

Based on the model of the ATP-RRM complex (Fig. 3), both the aromatic ring of adenine and triphosphate chain of ATP

appear to contribute to the interaction with the FUS RRM residues. To confirm this observation, we further conducted NMR HSQC titrations with AMP and PPP, as adenosine has a low solubility and consequently it was unable to reach high concentrations required for interacting with the FUS RRM domain. Interestingly, AMP was also able to induce large shifts of a small set of HSQC peaks of the FUS RRM domain (Supplementary Fig. 2). Detailed analysis revealed that AMP has a similar pattern of CSD (Fig. 4a) as ATP (Fig. 2b) except for that the HSQC peaks of Thr313, Asn314, and Thr338 no longer have large shifts. This result is completely consistent with the docking model of the ATP-RRM complex in which triphosphate chain directly interacts with Thr313, Asn314, and Thr338 (Fig. 3b). As such, the removal of two phosphate groups results in the loss of interaction with Thr313, Asn314, and Thr338 residues of the FUS RRM domain (Fig. 4b). On the other hand, we increased the concentration of AMP to 60 mM and subsequently obtained the dissociation constants of 7 residues with the HSQC CSD values >0.085, which range from 14.84 mM of Arg371 to 21.82 mM of Arg328 (Fig. 4c), with the average Kd value of 17.24 ± 2.61 mM, showing that compared with ATP, AMP has ~5-fold reduction of the binding affinity to the FUS RRM domain.

Furthermore, we also conducted the titrations with PPP and the results showed that it only induces large shifts of HSQC peaks of four residues including Thr313, Asn314, Met321, and Thr370 with their HSQC CSD > 0.06 (Fig. 5a and Supplementary Fig. 3). The Kd values of the four residues were fitted out to range from 8.25 mM of Met321 to 16.82 mM of Thr370 (Fig. 5b), with the average value of 12.63 ± 3.53 mM, implying ~3-fold reduction of affinity to the FUS RRM domain as compared to ATP. This result is also in a general agreement with the docking model in which the aromatic ring provides the direct contact with Thr326 and Arg328. On the other hand, it appears that, due to largely reduced molecular volume or/and binding affinity as compared to ATP, the isolated PPP has a slight re-orientation of the binding. Consequently PPP induces no meaningful shift of Asn284, Asn285, and Thr338 but triggers the large shift of a new residue Met321 with its HSQC CSD > 0.06 (Fig. 5c).

NMR HSQC titration results with ATP, AMP, and PPP support the docking model of the ATP-RRM complex, indicating that ATP specifically binds the FUS RRM domain. Furthermore, the relatively high binding affinity of ATP requires the presence of both adenine and triphosphate chain. The isolated AMP and PPP not only have much lower binding affinity but also have the reduced contacting surface on the FUS RRM domain. However, as judged from the fact that the average Kd value of ATP is much larger than the product of the average Kd values of AMP and PP, it appears that the binding events of adenine and triphosphate chain to the FUS RRM may not be independent[23].

**ATP- and nucleic acid-binding sites overlap.** So far, no NMR or crystal structure has been successfully determined for the FUS RRM domain in complex with any nucleic acids. However, it is well established that the RRM domains of hnRNPs share the well-conserved binding interfaces with nucleic acids[11–13], as demonstrated by the crystal structures of the TDP-43 RRM1 (Supplementary Fig. 4A) and RRM2 (Supplementary Fig. 4B) domains in complex with different ssDNA[24,25]. Interestingly, ATP binds to the same side of the FUS RRM domain (Fig. 3) and with a portion of binding surface even overlapping with the site binding ssDNA molecules. Furthermore, the CSD pattern of the FUS RRM domain induced by binding ATP here (Fig. 2b) shares a high similarity to those induced by binding RNA and ssDNA previously reported[13]. Briefly, RNA and ssDNA also triggered the

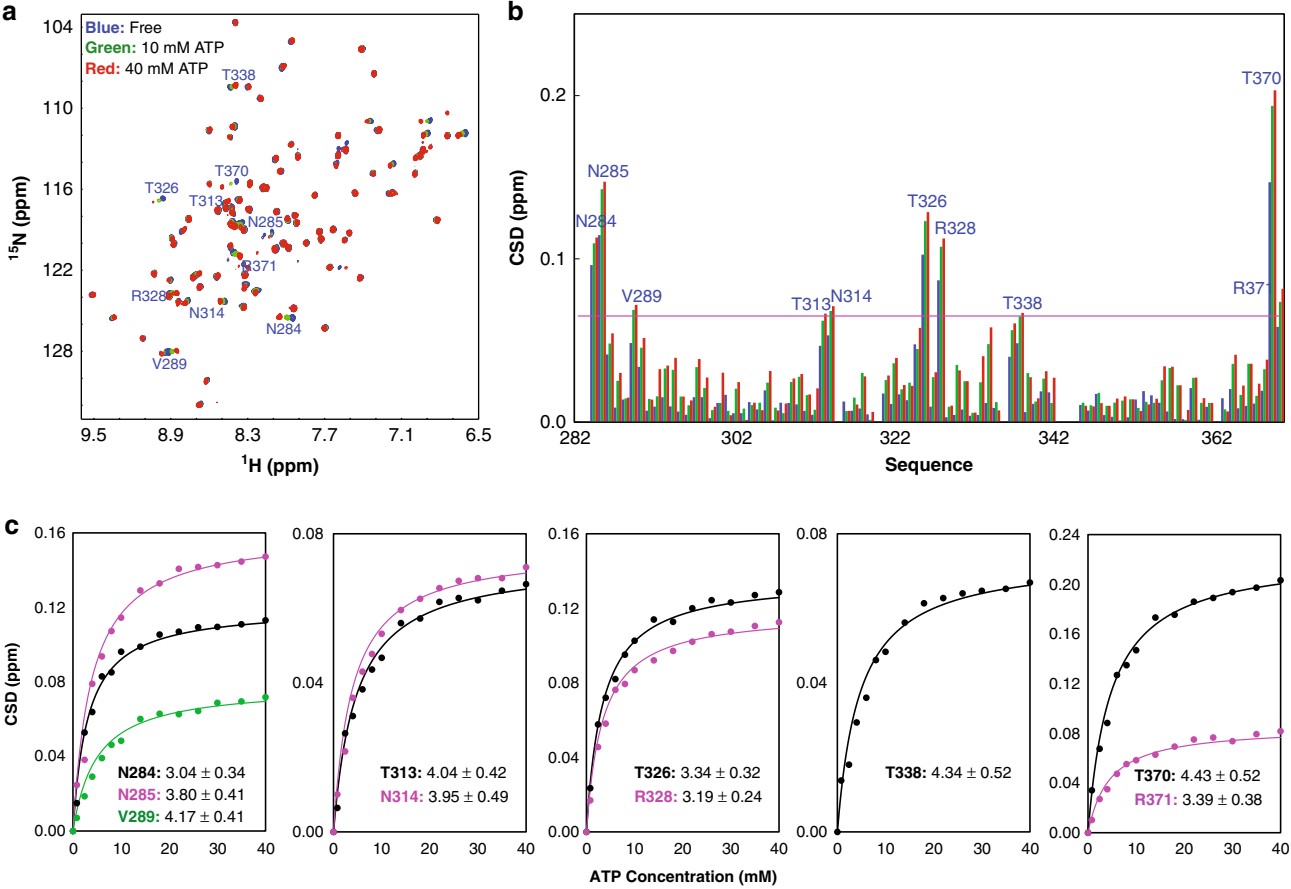

**Fig. 2** Adenosine triphosphate (ATP) specifically binds FUS RNA-recognition motif (RRM). **a** $^1$H-$^{15}$N nuclear magnetic resonance (NMR) heteronuclear single quantum coherence spectra of the $^{15}$N-labeled FUS RRM in the absence and in the presence of ATP at different concentrations. The highly shifted peaks are labeled. **b** Residue-specific chemical shift difference (CSD) of FUS RRM in the presence of ATP at 10 mM (blue), 30 mM (green), and 40 mM (red). Highly shifted residues are labeled, which are defined as those with the CSD values at 40 mM ATP > 0.065 (average value + one standard deviation) (purple line). **c** Fitting of 10 residue-specific dissociation constant (Kd): experimental (dots) and fitted (lines) values for the CSDs induced by addition of ATP at different concentrations. NMR titration experiments presented here were performed once

similar CSD patterns with highly shifted residues located over the same five regions of the FUS RRM domain as induced by ATP here, although more residues had highly shifted HSQC peaks upon binding nucleic acids[13].

Therefore, we asked a question whether the pre-binding of ATP to the FUS RRM domain will perturb its binding to nucleic acids. To address this, we selected a functional ligand, namely the 24-mer TssDNA, whose binding to the FUS RRM domain has been previously characterized to be an intermediate exchange process in the NMR timescale[13]. For this binding process, the gradual addition of TssDNA induced the large broadening of HSQC peaks and consequently led to the disappearance of HSQC peaks at high TssDNA concentrations[13]. As such, if the pre-binding of ATP with the FUS RRM domain largely enhances the binding of the FUS RRM domain to TssDNA, the binding process is expected to shift to a slow exchange process with two set of HSQC peaks: one from the free state and another from the complexed state. By contrast, if the pre-binding of ATP largely reduces the binding of the FUS RRM domain to TssDNA, the binding process is anticipated to shift to a fast exchange process with the HSQC peaks become gradually shifted upon stepwise addition of TssDNA as we currently observed on the binding process between the FUS RRM domain and ATP (Fig. 2a).

Here we first acquired HSQC spectra of the $^{15}$N-labeled FUS RRM domain titrated with TssDNA at molar ratios of 1:0.25, 1:0.5, 1:1, 1:2.5, 1:5, and 1:10 (RRM:TssDNA). Indeed, addition of

TssDNA induced dramatic broadening of HSQC peaks, and at the ratio of 1:10, almost all backbone HSQC peaks disappeared owing to the broadening (Fig. 6a). Subsequently, we prepared another sample of the $^{15}$N-labeled FUS RRM domain under the same conditions except for an extra addition of ATP to 3 mM, which mimic the ATP concentration in neurons (~3 mM), which was titrated with TssDNA at the same six ratios. The obtained spectra with ATP at 3 mM (Fig. 6b) showed no large difference from those without ATP (Fig. 6a). The results indicated that the presence of ATP failed to shift the binding event with TssDNA into either slow or fact exchange process, thus suggesting that the pre-binding of ATP would not dramatically perturb the binding process of the FUS RRM domain to its functional ligand TssDNA. This is reasonable as the Kd of the binding of the FUS RRM domain to ssDNA is ~23 μM[13]. In other words, the binding affinity of the FUS RRM domain to TssDNA is ~164 times higher than that to ATP.

We also attempted to explore whether ATP can bind other well-folded proteins. To assess this, we collected HSQC spectra of the $^{15}$N-labeled human profilin-1 (PFN1) titrated with ATP at different concentrations (Supplementary Fig. 4C). PFN1 is an 140 residue small protein that regulates actin polymerization, and a list of mutations have been identified to cause ALS[26–28]. It adopts a well-structured globular fold[27], in which a seven stranded β-sheet is sandwiched by N- and C-terminal α-helices on one face of the sheet and three small helical regions on the opposite face

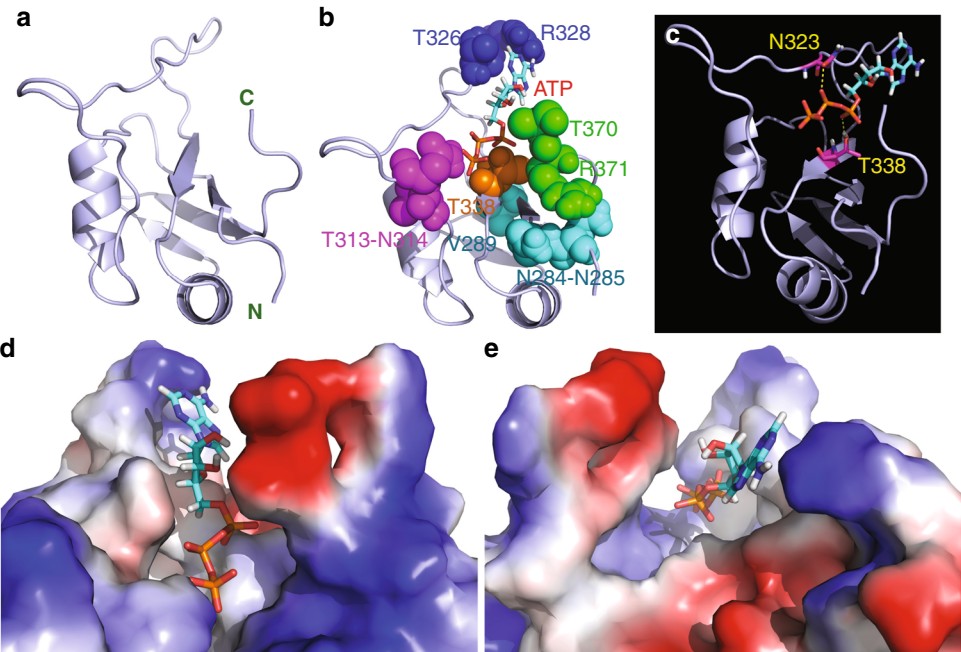

**Fig. 3** Three-dimensional model of the adenosine triphosphate (ATP)-RNA-recognition motif (RRM) complex. **a** The ribbon structure of the FUS RRM domain of the ATP-RRM complex. **b** The lowest energy docking model of the ATP-RRM complex. The structure of FUS RRM is displayed in ribbon, while ATP is in sticks. The ten residues with large CSD values are displayed in spheres and labeled. **c** The ATP-RRM complex model showing two hydrogen bonds (in yellow dotted lines) between triphosphate chain of ATP and side chains of Asn323 and Thr338 of FUS RRM respectively. **d–e** The ATP-RRM complex model with the RRM structure displayed in the electrostatic potential surface and ATP in sticks

(Supplementary Fig. 4D). Previously, we have conducted extensive NMR studies on PFN1 and its ALS-causing mutants[28]. As shown in Supplementary Fig. 4C, even with ATP concentrations reaching up to 20 mM, only two HSQC peaks from the exposed residues H120 and Gly121 (Supplementary Fig. 4D) showed some shifts. This result indicates that not all well-folded proteins have the specific ATP-binding pocket as identified on the FUS RRM domain.

**ATP kinetically inhibits amyloid formation**. The FUS RRM domain is required for its cytoxicity and indeed we previously found that it could spontaneously self-assemble into amyloid fibrils[14]. Therefore, here we aimed to assess whether ATP binding has effect on the fibrillation of the FUS RRM domain. Previously, we monitored the process by circular dichroism (CD), fluorescence, and NMR spectroscopy, as well as electron microscopy (EM). However, as the presence of ATP in the sample gave rise to large non-specific noise over both far- and near-ultraviolet (near-UV) CD regions. Therefore, here we monitored the formation of amyloid fibrils by the FUS RRM domain by Thioflavin-T (ThT) binding induced fluorescence, NMR spectroscopy, microscopy, and EM under the same conditions except for extra addition of ATP at different concentrations. Previously, without the presence of ATP, only after 2 days, almost all HSQC peaks of the FUS RRM become largely broadened and after 4 days completely disappeared; and the sample also become cloudy. On examination by microscopy, some condenses were formed that were further identified to be amyloid fibrils by EM[14]. Interestingly, here in the presence of ATP at 3 mM, even after 15 days NMR HSQC peaks showed no detectable disappearance only with minor shifts of several peaks (Supplementary Fig. 5A), and no ThT-binding induced fluorescence was detected (Supplementary Fig. 5B). Furthermore, the sample remained transparent, and no condense was observed as checked by microscopy. Furthermore, no

amyloid fibrils were formed as imaged by EM (Supplementary Fig. 5C). This result indicates that the presence of ATP inhibits the FUS RRM domain to self-assemble into amyloid fibrils.

We thus attempted to understand whether inhibition of amyloid formation is due to the enhancement of thermodynamic stability. Previously, we have characterized the thermal unfolding of FUS RRM under different conditions as monitored by CD and fluorescence spectroscopy. The melting temperatures were determined to be ~52–56 °C that is dependent on the probes (CD or fluorescence) used. As the presence of ATP resulted in large noises in both far- and near-UV CD regions, we planned to use the intrinsic UV fluorescence from residue Trp353 to report the thermal unfolding. Unexpectedly, we found that the addition of ATP led to dramatic quenching of the intrinsic UV fluorescence, and at 3 mM of ATP, the intrinsic UV fluorescence became very weak (Fig. 7a). We thus measured the Trp intrinsic UV fluorescence of several well-folded proteins, including PFN1 and TDP-43 RRM domains, and found that universally ATP could almost completely quench their intrinsic UV fluorescence at concentrations >4 mM. As such, we decided to monitor the thermal unfolding by a very popular method called differential scanning fluorimetry (DSF). Briefly, the thermal unfolding transition of a protein in the presence of a fluorescent dye SYPRO Orange is utilized to reflect the thermodynamic stability of a protein[29]. This dye with its fluorescence quenched in an aqueous solution becomes highly fluorescent in non-polar environments. As such, when a protein becomes unfolded, this dye will bind to the exposed hydrophobic patches of the unfolded intermediates, thus leading to a dramatic increase in visible fluorescence[29].

Figure 7b presents the melting curves of the FUS RRM domain in the presence of ATP at different concentrations, showing that the unfolding of the RRM domain by increasing temperatures is accompanied by increase in SYPRO Orange fluorescence. The melting temperature (Tm) values were determined by

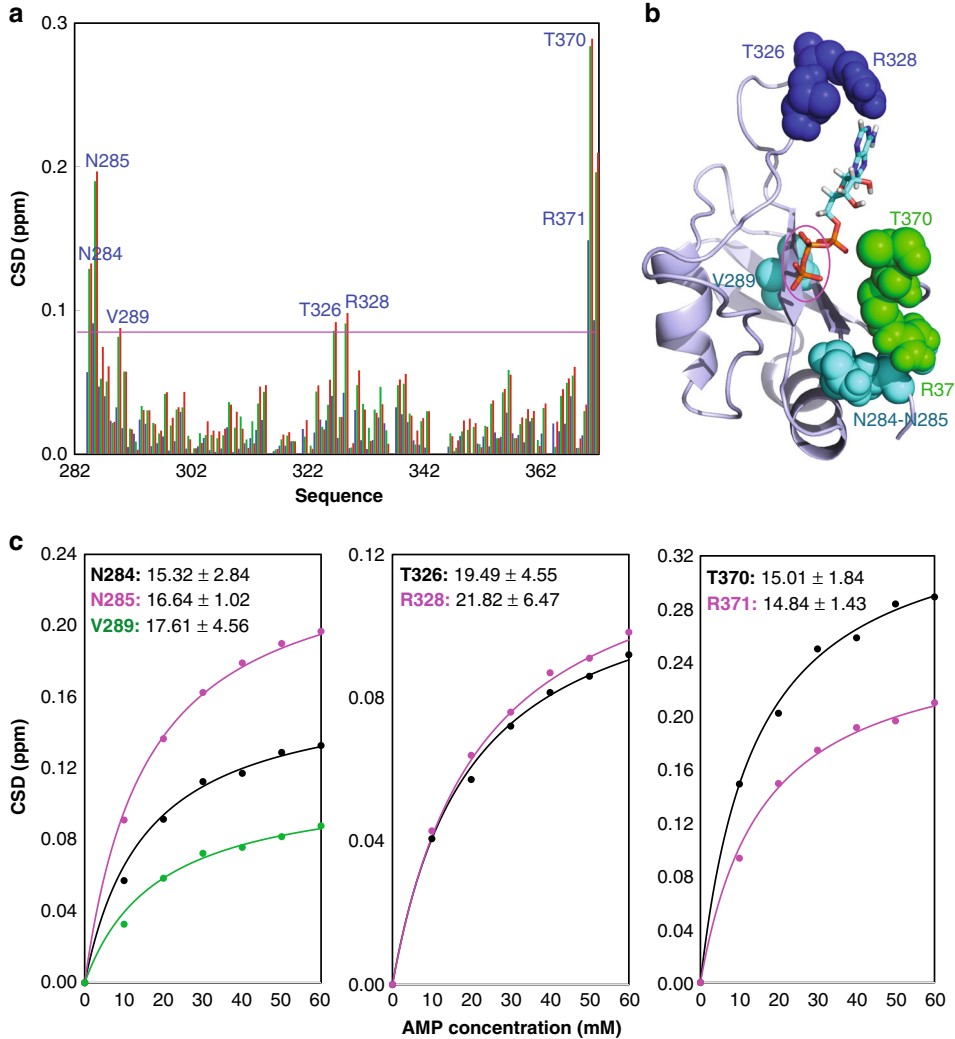

**Fig. 4** AMP binds FUS RNA-recognition motif (RRM) at a low affinity. **a** Residue-specific chemical shift difference (CSD) of FUS RRM in the presence of AMP at 10 mM (blue), 50 mM (green), and 60 mM (red). Largely shifted residues are labeled, which are defined as those with the CSD values at 60 mM ATP > 0.085 (average value + one standard deviation) (purple line). **b** The structure of the adenosine triphosphate (ATP)-RRM complex model in which only the residues largely perturbed by AMP are displayed in spheres. The pink cycle is utilized to indicate two phosphate groups of ATP but absent in AMP. **c** Fitting of seven residue-specific dissociation constant (Kd): experimental (dots) and fitted (lines) values for the CSDs induced by addition of AMP at different concentrations. The nuclear magnetic resonance titration experiments presented here were performed once

plotting the first derivative of the fluorescence emission as a function of temperature ($-dF/dT$). The FUS RRM domain in the absence of ATP has a Tm of ~55 °C, consistent with our previous results determined by CD and fluorescence probes[14]. Interestingly, the presence of ATP with concentrations up to 20 mM does not have any detectable effect on Tm values of the FUS RRM domain, suggesting that ATP binding has no large alteration of the thermodynamic stability. On the other hand, however, it is interesting to note that the $dF/dT$ values at Tm increased upon increasing ATP concentrations, implying that the hydrophobic patches exposed during unfolding may have different properties or/and degrees in the presence of ATP at different concentrations. We have also assessed the thermal unfolding of the full-length FUS with ATP at different concentrations but the obtained curves appeared to be very complex (Supplementary Fig. 5D). The full-length FUS additionally contains a folded zinc finger with ~30 residues and particularly a large portion (~75%) of the intrinsically disordered regions (Fig. 1b). While the new transition at ~38 °C might result from the unfolding of the folded zinc finger, the transition of the RRM domain was covered by a

large increase of $-dF/dT$ over 50–60 °C. This increase might be due to the temperature-dependent binding/release of the dye interacting with the hydrophobic patches over the intrinsically disordered region. In particular, the PLD over residues 1–165 contains many hydrophobic/aromatic residues, which are accessible for interacting with the dye even in the native state. As such, it is almost impossible to interpret the exact effect of ATP on the thermal stability of the full-length FUS.

## Discussion

Previously, ATP has been exhaustively characterized for its interactions with a myriad of proteins/enzymes, which are associated with the ATP-dependent energy transfers and reactions[1–3]. Only recently, ATP was identified to act as a biological hydrotrope to critically maintain protein homeostasis in cells but the underlying mechanisms remain largely unknown[2,3]. Very recently, we found that ATP can also behave as a bivalent binder to induce and subsequently dissolve LLPS of the intrinsically disordered C-terminal domain of FUS[4]. In the present study, we

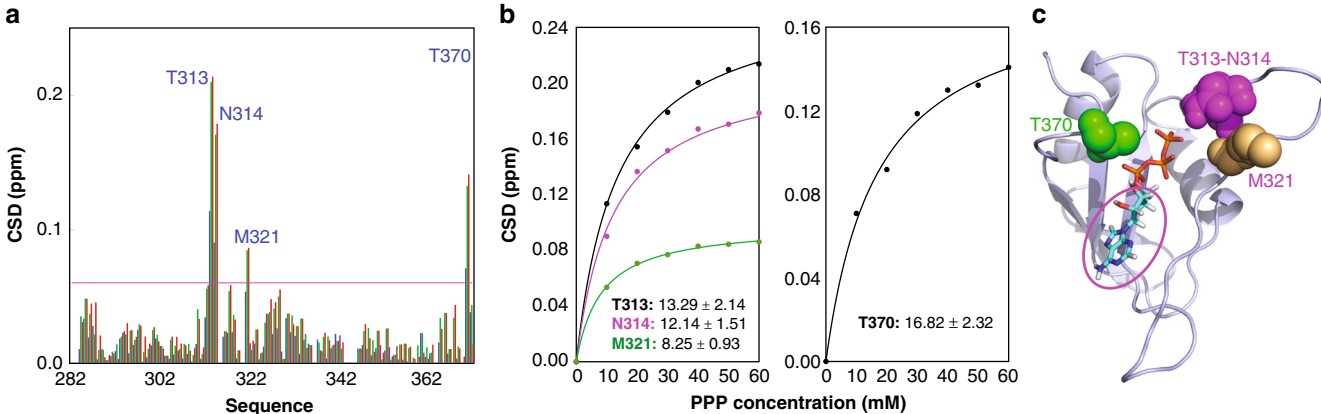

**Fig. 5** Triphosphate acid (PPP) binds FUS RNA-recognition motif (RRM) at a low affinity. **a** Residue-specific chemical shift difference (CSD) of FUS RRM in the presence of PPP at 10 mM (blue), 50 mM (green), and 60 mM (red). Largely shifted residues are labeled, which are defined as those with the CSD values at 60 mM PPP > 0.06 (average value + one standard deviation) (purple line). **b** Fitting of four residue-specific dissociation constant ($K_d$): experimental (dots) and fitted (lines) values for the CSDs induced by addition of PPP at different concentrations. **c** The structure of the adenosine triphosphate (ATP)-RRM complex model in which only the residues largely perturbed by PPP are displayed in spheres. The pink cycle is utilized to indicate adenosine of ATP but absent in PPP. The nuclear magnetic resonance titration experiments presented here were performed once

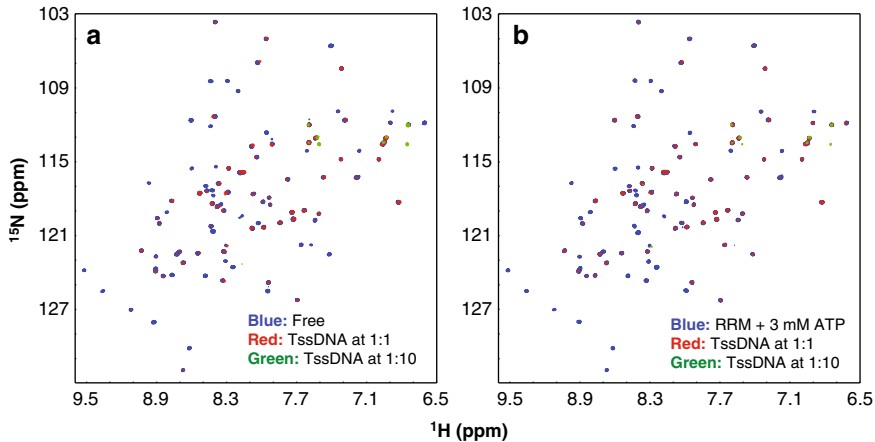

**Fig. 6** Adenosine triphosphate (ATP) has no large perturbation on the binding of RNA-recognition motif (RRM) with telomeric single-stranded DNA (TssDNA). **a** Heteronuclear single quantum coherence (HSQC) spectra of FUS RRM upon addition of 24-mer human TssDNA at different ratios. **b** HSQC spectra of FUS RRM with the pre-existence of 3 mM ATP upon addition of TssDNA at different ratios. The nuclear magnetic resonance titration experiments presented here were performed once

identified ATP to specifically bind to the well-folded FUS RRM domain at physiologically relevant concentrations, which inhibits the fibrillation of the RRM domain, although the binding has no detectable effect on the thermodynamic stability of the RRM domain. So what is the possible mechanism?

As shown in Fig. 3a, the FUS RRM domain contains a central four-stranded β-sheet sandwiched by two α-helices on the one side, as well as the N- and C-termini and loops on the other side. Previously, we have extensively characterized both thermodynamic and NMR dynamic stabilities of the FUS RRM domain and found that it has relatively high backbone dynamics even on the ps–ns timescale[14]. Therefore, we proposed that the self-assembly of the FUS RRM domain into amyloid fibrils is likely initiated by the dynamic opening of the structure to allow the inter-molecular oligomerization driven by the central β-sheets. As such, the fibrillation process of the FUS RRM domain in the free state has a relatively low kinetic barrier (Fig. 7c). Interestingly, the ATP-binding pocket of the RRM domain is mostly constituted by the N- and C-termini and loops (Fig. 3b). As a consequence, the binding of ATP into the pocket is expected to

attenuate the structure opening to certain degree, thus resulting in the increase of the kinetic barrier for the fibrillation (Fig. 7c).

Strikingly, FUS represents a member of a large family of RNA-binding proteins (RBPs) including TDP-43 whose aggregation/fibrillation is universally associated with various neurodegenerative diseases[30], which contain at least one RRM domain. As the RRM domains of this family of RBPs have very similar structures, as well as conserved surfaces for binding various nucleic acids, it is likely that ATP may bind at least a large portion of these RRM domains. Indeed, our preliminary studies indicate that ATP could also bind to the RRM domains of TDP-43 we previously studied[31]. In the future, it is of significant interest to explore whether the binding of ATP to other RRM domains will also inhibit their pathological fibrillization.

It has been well established that the amyloid fibrillation is critically associated with "gain of toxicity" for human proteins involved in various neurodegenerative diseases[2,3,7,8,26,27,30,32–44]. Indeed, the aggregation/fibrillation of FUS and TDP-43 has been extensively demonstrated to be involved in "gain-of-toxicity" for ALS/FTD pathogenesis[6–8,35–44]. Intriguingly, however, although

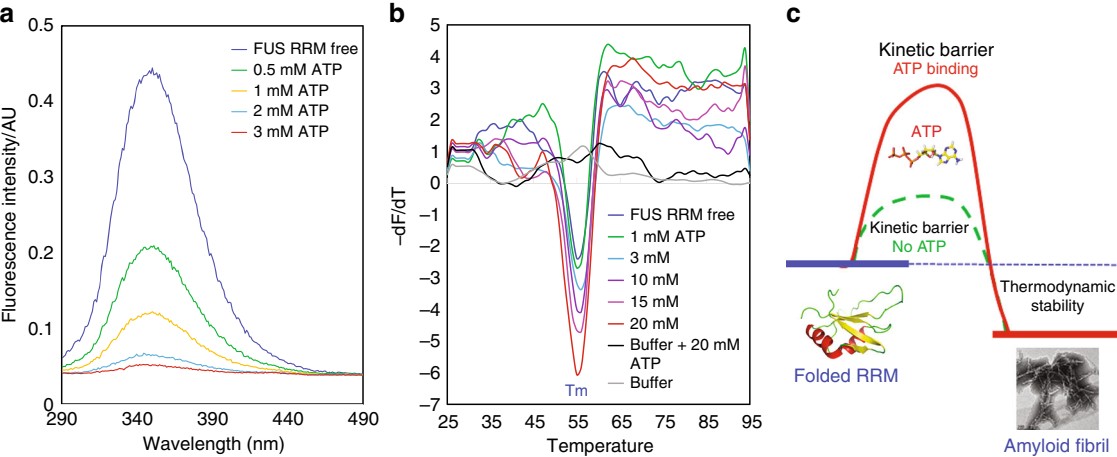

**Fig. 7** Adenosine triphosphate (ATP) has no detectable effect on thermodynamic stability of RNA-recognition motif (RRM). **a** Intrinsic ultraviolet (UV) fluorescence spectra from residue Trp353 of FUS RRM in the presence of ATP at different concentrations. The fluorescence intensity was reported in arbitrary unit. **b** Differential scanning fluorimetric melting curves of thermal unfolding of FUS RRM in the presence of ATP at different concentrations by plotting the first derivative of the fluorescence emission as a function of temperature ($-dF/dT$). Here the Tm is represented as the lowest point of the curve. **c** A proposed diagram to illustrate that the specific ATP binding kinetically inhibit the self-assembly into amyloid fibrils without large alteration of the thermodynamic stability of FUS RRM. These data represent one out of three experiment repeats with similar results

the N-terminal PLD of FUS over residue 1–165 is sufficient to rapidly undergo the amyloid fibrillization[35–39], its RRM domain together with RGG2 is further needed to manifest cytotoxicity[8]. As RGG2 and RGG1 have highly similar sequence properties (Fig. 1a), which are incapable of forming amyloid fibrils by themselves[35–39], the fibrillation of the RRM domain thus appears to play a key role in "gain-of-toxicity" of FUS. Recent studies might provide a possible mechanism for this observation: unlike the classic irreversible amyloid fibrils stabilized by strong hydrophobic interactions[32–34], the fibril formed by the FUS PLD is only stabilized by relatively weak polar interactions and thus reversible[39]. Therefore, it appears that the RRM fibrillization driven by the hydrophobic interaction is critically needed to convert the reversible non-toxic form into irreversible toxic form of the FUS fibril, as extensively observed on the classic irreversible amyloid fibrils including those formed by TDP-43 RRM domains[32–34,42].

FUS molecules are usually localized in the nucleus but under the stress conditions might be re-localized into the cytoplasm to form stress granules by LLPS, which is now recognized to be a common principle for forming a variety of membrane-less organelles[44–46]. However, upon phase separation, the local protein concentrations within the liquid droplets could increase by ~50–100 times as compared to that in the surrounding solution[45–47]. Therefore, LLPS is widely thought to also promote the transition from the dynamic liquid droplets into inclusion/fibrils responsible for neurodegenerative diseases[35–47]. For example, the accumulation of FUS inclusions in the cytoplasm is the hallmark for ALS/FTD. Recently, the cellular concentration of FUS was estimated to be ~12 μM, but in the nucleus LLPS of FUS is largely suppressed by the presence of RNA at high concentrations[47]. However, owing to the low concentrations of RNA, in the cytoplasm FUS will phase separate into the liquid droplets, which may further transit into irreversible aggregation/fibrillation under long stress time or/and pathological conditions. This thus explains the long-standing observation that the re-localization or mis-localization of FUS into the cytoplasm might lead to its severe aggregation/fibrillation characterized by ALS/FTD[47].

LLPS of FUS is functionally essential for forming reversible stress granules in the cytoplasm. On the other hand, the formation of the liquid droplets will unavoidably promote the further

transition into the irreversible and pathological aggregation/fibrillation. So the mechanism to allow the reversible LLPS of FUS but to inhibit the further transition into the irreversible aggregation/fibrillation is critical to prevent FUS from "gain-of-toxicity" in the cytoplasm. Therefore, the binding of ATP to the RRM domain may represent such a safeguard mechanism, because ATP has high concentrations in the cytoplasm. Interestingly, out of different types of human cells, neuron cells were found to have relatively low ATP concentrations (~3 mM)[1–3]. This fact may explain why FUS and other RBPs such as TDP-43 in the same family are particularly prone to aggregation in the cytoplasm of neurons. Furthermore, it is well known that, upon ageing, the ATP concentrations in human cells including neurons are reduced. This reduction might partly account for the observation that the risk of neurodegenerative diseases including ALS/FTD increases with age.

## Methods

**Preparation of recombinant FUS and profilin-1 proteins**. The same protocols we previously established were used in the present study to express and purify the human full-length FUS, its RRM domain[14], and profilin-1[28]. To generate isotope-labeled proteins for NMR studies, the bacteria were grown in M9 medium with addition of $(^{15}NH_4)_2SO_4$ for $^{15}N$-labeling[14,28]. The protein concentrations were determined by the UV spectroscopic method in the presence of 8 M urea, under which the extinct coefficient at 280 nm of a protein can be calculated by adding up the contribution of Trp, Tyr, and Cys residues[14,28,48].

ATP, AMP, adenosine, and PPP were purchased from Sigma-Aldrich with the same catalog numbers as previously reported[2]. MgCl$_2$ was added into ATP for stabilization by forming the ATP-Mg complex[2]. The 24-mer telomeric ssDNA (TssDNA) with a sequence of (TTAGGG)$_4$ were purchased from a local company[49]. The fluorescent dye SYPRO Orange (S5692–50UL) was purchased from Sigma-Aldrich. The protein samples, as well as ATP, AMP, PPP, adenosine, and TssDNA, were all prepared in 10 mM sodium phosphate buffer containing 150 mM NaCl with a final pH of 6.8.

**NMR characterizations**. All NMR experiments were acquired at 25 °C on an 800 MHz Bruker Avance spectrometer equipped with pulse field gradient units and a shielded cryoprobe as described previously[14,28,48]. For NMR HSQC titration studies of the interactions of RRM/PFN1 with ATP, AMP, adenosine, PPP, or TssDNA, two-dimensional $^1H$-$^{15}N$ NMR HSQC spectra were collected on the $^{15}N$-labeled FUS RRM domain or PFN1 samples at a protein concentration of 40 μM in 10 mM sodium phosphate buffer containing 150 mM NaCl (pH 6.8) at 25 °C in the presence of ATP, AMP, adenosine, PPP, or TssDNA at different concentrations as specified in the "Results" section. NMR spectra were processed with NMRPipe[50] and analyzed with NMRView[51].

**Calculation of CSD and data fitting to obtain Kd**. To calculate CSD, the HSQC spectra were superimposed for the $^{15}$N-labeled FUS RRM domain collected in the free state and in the presence of ATP, AMP, or PPP at different concentrations. Subsequently, the shifted HSQC peaks could be identified and further assigned to the corresponding RRM residues based on the sequential assignment we previously obtained[14]. The CSD was calculated by an integrated index calculated by the following formula:

$$\mathrm{CSD} = \left( \left( \Delta^1 H \right)^2 + \left( \Delta^{15} N \right)^2 / 4 \right)^{1/2}.$$

In order to obtain residue-specific dissociation constant (Kd), we fitted the shift traces of the residues with large shifts of HSQC peaks (CSD > average + STD) by using the one binding site model[15–17,20] with the following formula:

$$\mathrm{CSD_{obs}} = \mathrm{CSD_{max}} \left\{ ([P] + [L] + (Kd)) - \left[ [P] + [L] + (Kd)^2 - [P][L] \right]^{1/2} \right\} / 2[P]$$

Here [P] and [L] are molar concentrations of FUS RRM and ligands (ATP, AMP, or PPP), respectively.

**Molecular docking**. The structure model of the ATP-RRM complex was constructed by use of the HADDOCK software[19,20] in combination with crystallography and NMR system (CNS)[52], which makes use of CSD data to derive the docking that allows various degrees of flexibility. The CNS file of ATP molecule was obtained from PRODRG server[53]. Briefly, HADDOCK docking procedure for the ATP-RRM complex was performed in three stages: (1) randomization and rigid body docking; (2) semi-flexible simulated annealing; and (3) flexible explicit solvent refinement. The ATP-RRM structure with the lowest energy score was selected for the detailed analysis and display by Pymol (The PyMOL Molecular Graphics System, Version 0.99rc6 Schrödinger, LLC).

**Fluorescence spectral measurements**. To assess the ATP-induced quench of the intrinsic Trp UV fluorescence of the FUS RRM domain, the spectra were measured with the excitation wavelength at 280 nm at 25 °C with a RF-5301 PC spectrophotometer (Shimadzu, Japan) on the sample of FUS RRM at 40 μM in 10 mM sodium phosphate buffer containing 150 mM NaCl (pH 6.8) in the presence of ATP at different concentrations.

ThT-binding assay followed the same protocol we previously used to monitor amyloid formation of FUS RRM[14]. Briefly, a 2-mM ThT stock solution was prepared by dissolving ThT in milli-Q water and filtered through a 0.22-μm Millipore filter. The fresh working solution was prepared by diluting the stock solution into 10 mM sodium phosphate buffer containing 150 mM NaCl (pH 6.8) to reach a final ThT concentration of 50 μM. A 10-μL aliquot of each incubation solution or 10 μL aliquot of the incubation buffer as the control, was mixed with 130 μL of the ThT working solution in the dark for 10 min. The fluorescence emission spectra were acquired for three repeats with the excitation wavelength at 442 nm and slit widths: excitation at 5 nm and emission at 10 nm[14].

**Differential interference contrast (DIC) and EM imaging**. The samples of FUS RRM for assessing the formation of amyloid fibrils were prepared as previously reported[14], except for additional addition of ATP at different concentrations. The incubation samples were checked every day by differential interference contrast (DIC) and imaged at different time points by EM[14]. Briefly, aliquot of 40 μL of incubation samples was used for DIC check every day by a DIC microscopy (OLYMPUS IX73 Inverted Microscope System with OLYMPUS DP74 Color Camera) or imaged by a TEM microscopy (Jeol Jem 2010f Hrtem, Japan) operating at an accelerating voltage of 200 kV as we previously described[14,41].

**Determination of thermodynamic stability by DSF**. DSF was used to determine the thermodynamic stability of the FUS RRM domain in 10 mM sodium phosphate buffer containing 150 mM NaCl (pH 6.8) in the presence of ATP at different concentrations. DSF experiments were performed using the CFX384 Touch™ Real-Time PCR Detection System from BIO-RAD, following the SYBR green melting protocol to obtain Tm value[29]. Briefly, in a single well of a 384-well PCR plate, a 10-μL reaction solution was placed, which contains the FUS RRM domain at 10 μM, ATP at different concentrations, and 10× SYPRO Orange in 10 mM sodium phosphate buffer containing 150 mM NaCl (pH 6.8). Plates were sealed with a quantitative PCR adhesive optical seal sheet (Microseal 'B' Adhesive Sealing Films, BIO-RAD) and then spun at 1000 rpm for 1 min to remove bubbles. The program in Real-Time PCR instrument was set to SYBR green and ran the temperature scan from 25 °C to 95 °C with the increment of 1 °C/min. Upon completion, the obtained thermal unfolding curves were displayed as the first derivatives (dF/dT) by the reverse transcriptase PCR software Bio-Rad CFX Manager 3.0.

**Statistics and reproducibility**. For NMR experiments, the exploratory HSQC experiments titrated with ATP, AMP, PPP, and ssDNA at different concentrations were first conducted to identify the optimized concentration ranges of ATP, AMP, PPP, and ssDNA, which could give the reliable fitting of Kd values. The final HSQC titrations spectra used for fitting of Kd values were performed once with the optimized points of ATP, AMP, PPP and ssDNA concentrations.

The measurements of thermodynamic stability and ThT-binding induced fluorescence were performed with three repeats.

**Reporting summary**. Further information on research design is available in the Nature Research Reporting Summary linked to this article.

## Data availability

The data supporting the findings of this study are available within the paper and its supplementary information files.

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

## Acknowledgements

This study is supported by the Ministry of Education of Singapore (MOE) Tier 2 Grant MOE2015-T2–1–111 to J.S.

## Author contributions

J.S. and J.K. conceived and designed the experiments. All authors performed the research, analyzed the data, and wrote and reviewed the manuscript.
