## [Peer Review File · Communications Biology]

Referee expertise:

Referee #1: FUS

Referee #2: phase transitions, neurodegeneration

Referee #3: NMR, phase transitions

Reviewers' comments:

Reviewer #1 (Remarks to the Author):

Adenosine triphosphate (ATP) is well-known as the energy currency of a cell. However, recently there have been some remarkable discoveries of the potential role of ATP in cells such as a biological hydrotrope to maintain protein homeostasis. However, the underlying molecular mechanism underlying ATP's hydrotropic activity is unknown. In this manuscript, Kang et al. combine NMR techniques with a range of other biophysical techniques to elucidate the molecular interactions responsible for the hydrotropic activity of ATP using the RNA binding domain of the protein FUS. The results of this paper are very convincing and extremely exciting for the wider scientific community. I will strongly recommend the immediate publication of this manuscript after some recommended improvements.

1) A weakness of most of the NMR based papers on FUS is the experiments are mostly focussed on using only a part of the protein and not the full-length protein. This is understandable, owing to the limitations of using the full-length protein for NMR studies. However, it will be interesting to see the results for the thermodynamic stability experiments of ATP with full-length FUS in addition to the RRM only (figure 7A-B). I anticipate this will be a relatively easy and less time-consuming experiment for the authors.

2) For an enhanced clarity in figure 3, the authors should represent the results of the different interactions of amino acids with the ATP molecule in a much more simplified scheme. They should use the same scheme in figure 4 and 5, to demonstrate which are the interactions that cannot be achieved by AMP and PPP. They should also merge figure 4 and 5.

Reviewer #2 (Remarks to the Author):

In the manuscript titled "novel mechanism for ATP to specifically bind and inhibit fibrillization of ALS/FTD-causing FUS RRM" Kang et al describe their results on the structural changes in the FUS RRM domain upon binding of ATP.

I find the study of general interest and it has the potential to add to the current intensive investigations of phase separation and fibrillization of FUS, however, it also has some gaps in the text and figures etc. IN the following is a list of the specific points that the authors need to address before publication.

Point-by-point comments:

Title:

Remove "ALS-causing", this is not really known. Add "domain" in the end.

In the abstract:

- it is unclear what is TssDNA, please define
- Remove the last phrase: " which is involved in a large spectrum of neurodegenerative diseases"; you are not showing disease relevant data.

In the introduction section:

- Too many times the word "extremely"; who defines what is extreme in a cell.
- Some typos and unusual phrasing in the text; I suggest some editing of the text.
- Page 4: define for what the is known about the function of eth C-terminal domain.
- Page 5: give some examples of NMR in weak binding, e.g. in unfolded proteins and in liquid phase separation.

In the results section:

- Page 6: concentrations of 1-4 mM ATP are not "relatively low" but still high. Here and throughout the manuscript, I suggest to just state the actual concentration without words of judgement.
- Page 6: the authors state that they titrated 14 concentrations, but do not show the data. Please show them, or remove that number.
- How does 40 μ M FUS relate to other studies looking at FUS fibrilization and phase separation?
- Figure 1 legend: lane 4 says "(1)" but this number does not appear in the figure; I suggest to remove it from the legend
- Figure 2A: shifts in panel A are not visible at this magnification; please add zoom-ins for the shifted residues showing the actual amount of the shift
- Figure 2A: It seems that 40mM ATP shift the spectrum less than 10mM ATP compared to 0mM ATP; please explain and comment in text or discussion.
- Figure 2C: change the color coding of the traces so that they match Panel A; use green+pink instead of green+red to enable color-blind people reading the graphs
- Figure 3A: I suggest to reduce the size of the "balls" indicating the shifted residues to actually see some details; please add the lowest energy model of FUS RRM without ATP for comparison.
- Figure 3C: the yellow lines for the hydrogen bonds are not visible; please chose a dark color
- Page 9: check the order of figures and their citations in the text; I believe something got mixed up: citation for figure 6A+B comes before citation of Figure 4. Maybe shift TDP43 structures in Figure 6A+B into Supplemental data, since they are not immediately relevant for your results.
- Figure 4+5: Same as in Figure 2, the spectrum shown is too small and does not allow to see the shifts; please add zoom-ins for the shifted residues, and adjust trace color to match panels A.
- Figure 6A+B: where comes the data from? Add data or cite origin of data.
- Figure 6C: which FUS concentration was used? Also show zoom-ins of shifted regions for better visibility.
- Figure 6: in general, please check panel labeling and citations in the text; they do not match (I believe D should be E, and E should be F; panel D for FUS RRM with ATP+TssDNA is missing although explained in the text; please add the data in Figure.
- Page 10: at the bottom; the authors say that they measured spectra of PFN1 with up to 20mM ATP; please add the with higher ATP concentrations, at least up to 40mM ad for FUS RRM.
- Page 11: at the bottom; explain what is intrinsic UV fluorescence.
- Figure 7A: ATP only Control needs to be added to the graph.
- Figure 7B: ATP only control, and Dye only control are missing in the graphs: please add this data. Can the authors also add data melting curves of FUS RRM with ADP and AMP?

Discussion section:

The first half of the discussion repeats the results explained before and is thus redundant. Instead the authors should discuss their findings more in detail in the context of other previous

publications looking at the liquid-liquid phase separation and fibrillization of FUS and the role of the different FUS domains in the cell. What are the concentrations in the cell, when does FUS RRM plays a role, how could the detected binding of ATP in neurons be related to pathological FUS in ALS and FTD, etc?

Reviewer #3 (Remarks to the Author):

Kang et al. put forth the intriguing hypothesis that ATP binds specifically to folded RNA recognition motifs (RRMs) with affinity in the single digit millimolar, which is consistent with the physiological ATP concentration in the cell. The authors propose that ATP binding kinetically inhibits RRM-mediated fibrillization associated with neurodegenerative diseases. The authors move one step forward to propose that this mechanism is likely a general feature of other RRM-containing proteins that are involved in neurodegeneration.

The data pertaining to the structural and thermodynamic characterization of FUS-RRM:ATP complex, based on NMR, computational docking based on NMR-derived restraints and stability studies, are convincing. However, based on the data presented in this manuscript, the mechanistic conclusions seem to be overreaching. For that reason, I consider that the manuscript is not ready for publication yet. I do, however, recommend reconsideration of the manuscript after appropriate revisions are made, to address the concerns listed below.

1. Line 230-237: Please show the NMR data that supports the claims in this paragraph.
2. Line 238 and following paragraph: "We also asked the question whether ATP can bind all well-folded proteins." This question is way too broad sweeping. The profilin-1 NMR data showing that its folded domain does not bind ATP does not add any value to the paper, since it is exemplifying a single point, arguably the exception. I suggest the authors either remove or rethink this section.
3. Lines 257-268: Please include the supporting ThT fluorescence, NMR, microscopy and EM data supporting the claims presented in this paragraph.
4. Lines 318-320: Please discuss whether any residues from the identified FUS-RRM ATP binding site overlap with the residues within RRM that are associated with fibrillization.
5. Line 332: "[...] RRM domain is absolutely needed to manifest cytotoxicity (11)" This statement is inaccurate. In ref (11), fig. 3A, a FUS truncation containing the PLD and RRM domain (residues 1-373) was not sufficient to induce toxicity.
6. Regarding the relevance of the ATP-binding to FUS in reducing RRM-mediated fibrillization, please discuss any published evidence that addresses how often FUS exists in cells in an RNA-unbound state. What is the relevance of the mechanism if the protein is always bound to RNA, which competes off the ATP? Discuss how the data in this manuscript fits in the framework presented in Maharana et al, Science (2018), PMID: 29650702, that RNA buffers the phase separation of aggregation-prone proteins?

Point-to-point responses to the comments by three reviewers

First of all, I would like to thank three reviewers for the kind comments which act to significantly improve the revised manuscript.

To address all the comments, I had added new results as three reviewers kindly requested, as well as extensively revised the manuscript. Briefly, I added five supplementary figures, included new references, modified the figures to enhance visualization and conducted extensive revisions of the text.

The revised parts were colored in blue in the manuscript with changes tracked.

Reviewer #1 (Remarks to the Author):

1) A weakness of most of the NMR based papers on FUS is the experiments are mostly focussed on using only a part of the protein and not the full-length protein. This is understandable, owing to the limitations of using the full-length protein for NMR studies. However, it will be interesting to see the results for the thermodynamic stability experiments of ATP with full-length FUS in addition to the RRM only (figure 7A-B). I anticipate this will be a relatively easy and less time-consuming experiment for the authors.

Response: thanks so much for the kind suggestion. We have performed the experiments and the results were included in Figure S5D. The full-length FUS additionally contains the intrinsically-disordered N-terminal prion-like domain and the C-terminal RG regions, as well as the well-folded Zinc Finger. Therefore, the interactions of the FUS domains/regions with the dye during thermal unfolding appear to be extremely complex as shown. As the prion-like domain is intrinsically disordered but contains many aromatic/hydrophobic residues, the intrinsically-disordered regions in the full-length FUS might bind to the dye even in the native state, while at high temperatures; some bound dye might be released. Indeed, it seems that compared to the unfolding curves of RRM, in the curves of the full-length FUS, there is another unfolding transition at ~38 degree which might be from the Zinc Finger. Interestingly, there appears a significant release of the dye at ~58 degree, which might be from the significant release of the dye bound to the hydrophobic patches of the intrinsically-disordered regions. As such, the unfolding transition for RRM is covered. Therefore, it is almost impossible to interpret the changes of the curves of the full-length FUS in the presence of ATP at different concentrations.

I thus added a discussion as “We have also assessed the thermal unfolding of the full-length FUS with ATP at different concentrations but the obtained curves appeared to be very complex (Supplementary Figure 5D). The full-length FUS additionally contains a folded zinc finger with ~30 residues and particularly a large portion (~75%) of the intrinsically-disordered regions (Fig. 1B). While the new transition at ~38 °C might result from the unfolding of the folded zinc finger, the transition of the RRM domain was covered by a large increase of $-dF/dT$ over 50-60 °C. This increase might be due to the temperature-dependent binding/release of the dye interacting with the hydrophobic patches over the intrinsically-disordered region. In particular, the prion-like domain over residues 1-165 contains many hydrophobic/aromatic residues which are accessible for interacting with the dye even in the native state. As such, it is almost impossible to interpret the exact effect of ATP on the thermal stability of the full-length FUS.”

2) For an enhanced clarity in figure 3, the authors should represent the results of the different interactions of amino acids with the ATP molecule in a much more simplified scheme. They should use the same scheme in figure 4 and 5, to demonstrate which are the interactions that cannot be achieved by AMP and PPP. They should also merge figure 4 and 5.

Response: thanks so much for the kind suggestions. To address the comments,

1) As the second reviewer also has the same comment, I tried different ways and found it help by adding a simplified structure of the FUS RRM domain only. Furthermore, I also attempted to enhance visualization of the hydrogen bonds by changing the background of Figure 3C into black.

2) As kindly suggested, I have added the figures to demonstrate the interactions that cannot be achieved by AMP and PPP.

3) On the other hand, after inclusion of the sub-figures of the structures, I had to move NMR HSQC spectra for AMP and PPP respectively to Supplementary in order to allow good visualization.

4) I tried to merge Figures 4 and 5 but the merged figure was too small to allow good visualization. Consequently the results for AMP and PPP were still presented in two figures.

Reviewer #2 (Remarks to the Author):

Title: Remove “ALS-causing”, this is not really known. Add “domain” in the end.

Response: thanks so much and I have modified the title as kindly suggested.

In the abstract:

it is unclear what is TssDNA, please define

Response: thanks so much; and I have removed it in the abstract due to the 150-word constraints but added the detail in the main text.

Remove the last phrase: “which is involved in a large spectrum of neurodegenerative diseases”; you are not showing disease relevant data.

Response: thanks so much and I have removed it.

In the introduction section:

Too any times the word “extremely”; who defines what is extreme in a cell.

Response: thanks so much and I have removed them.

Some typos and unusual phrasing in the text; I suggest some editing of the text.

Response: thanks so much and I have conducted extensive editing.

Page 4: define for what the is known about the function of eth C-terminal domain.

Response: thanks so much and I added it with new references as “Very interestingly, the intrinsically disordered RG/RGG regions of FUS have been recently identified to bind various nucleic acids including DNA and RNA with degenerative sequence specificity (9,10).”.

Page 5: give some examples of NMR in weak binding, e.g. in unfolded proteins and in liquid phase separation.

Response: thanks so much and I added it with new references as “NMR spectroscopy is very powerful in not only detecting but also quantifying residue-specific parameters for the very weak binding events including the formation of “fuzzy complex” and liquid-liquid phase separation (LLPS) of FUS (15-18).”.

In the results section:

Page 6: concentrations of 1-4 mM ATP are not “relatively low” but still high. Here and throughout the manuscript, I suggest to just state the actual concentration without words of judgement.

Response: thanks so much for the comment, and I have modified them accordingly.

Page 6: the authors state that they titrated 14 concentrations, but do not show the data. Please show them, or remove that number.

Response: thanks so much for the comment. Although in the previous version, we did not present original HSQC spectra at 14 concentrations due to space limitation, we used the chemical shifts of all 14 concentrations for fitting K_d, as illustrated by 14 data in Figure 2C. Now as kindly suggested by the reviewer, we added the traces of significantly shifted HSQC peaks with more concentration points.

How does 40 uM FUS relate to other studies looking at FUS fibrilization and phase separation?

Response: thanks so much for the comment. Currently, only we previously studied the fibrillation of FUS RRM at the same concentrations. For the solid-state NMR study on the fibril structure formed by the prion-like domain, they used different but very high concentrations in order to generate the fibrils for NMR studies.

Figure 1 legend: lane 4 says “(1)” but this number does not appear in the figure; I suggest to remove it from the legend

Response: thanks so much and I have removed it.

Figure 2A: shifts in panel A are not visible at this magnification; please add zoom-ins for the shifted residues showing the actual amount of the shift

Response: thanks so much and I have included them as Supplementary Figure 1.

Figure 2A: It seems that 40mM ATP shift the spectrum less than 10mM ATP compared to 0mM ATP; please explain and comment in text or discussion.

Response: thanks so much for pointing out this mistake. This is due to a mistake in correlating ATP concentrations to the colors of HSQC spectra, and I have corrected them. In fact, red is for 40 mM ATP while green is for 10 mM ATP.

Figure 2C: change the color coding of the traces so that they match Panel A; use green+pink instead of green+red to enable color-blind people reading the graphs

Response: thanks so much for the suggestion and I have changed them.

Figure 3A: I suggest to reduce the size of the “balls” indicating the shifted residues to actually see some details; please add the lowest energy model of FUS RRM without ATP for comparison.

Response: thanks so much for this suggestion and I have added the RRM structure as Figure 3A. On the other hand, I have also played with the software to change the size but failed. The software does not allow change the size of the sphere as the size is defined within the software by the van der waals radius of an atom.

Figure 3C: the yellow lines for the hydrogen bonds are not visible; please chose a dark color

Response: thanks so much for this suggestion and I have attempted to change the color with the software but also failed. So I changed the background of Figure 3C into black to allow an enhanced visualization of the yellow dotted lines.

Page 9: check the order of figures and their citations in the text; I believe something got mixed up: citation for figure 6A+B comes before citation of Figure 4. Maybe shift TDP43 structures in Figure 6A+B into Supplemental data, since they are not immediately relevant for your results.

Response: thanks so much for this suggestion and I have corrected the orders. I also moved the two structures into Supplementary Figure 4.

Figure 4+5: Same as in Figure 2, the spectrum shown is too small and does not allow to see the shifts; please add zoom-ins for the shifted residues, and adjust trace color to match panels A.

Response: thanks so much for this comment. In the revised version, to allow better visualization I have moved the two HSQC spectra into Supplementary Figure S2 and S3 with

the inclusion of the expanded traces of significantly shifted HSQC peaks.

Figure 6A+B: where comes the data from? Add data or cite origin of data.

Response: thanks so much for this comment. In the revised version, I have moved the two structures into Supplementary Figure 4 and added the references.

Figure 6C: which FUS concentration was used? Also show zoom-ins of shifted regions for better visibility.

Response: The FUS concentration is 40 μ M, the same as used for ATP, AMP and PPP titrations.

Now I moved the structures of the TDP-43 RRM-ssDNA into Supplementary and it should be possible to visualize the spectra better. In fact, there is no significant shift but only extensive intensity reduction or disappearance of the HSQC peaks upon adding TssDNA.

Figure 6: in general, please check panel labeling and citations in the text; they do not match (I believe D should be E, and E should be F; panel D for FUS RRM with ATP+TssDNA is missing although explained in the text; please add the data in Figure.

Response: thanks so much for this comment, and I have corrected them and added the detail for TssDNA.

Page 10: at the bottom; the authors say that they measured spectra of PFN1 with up to 20mM ATP; please add the with higher ATP concentrations, at least up to 40mM ad for FUS RRM.

Response: thanks so much for this comment, and we have attempted to add ATP to higher concentration but PFN1 became precipitated.

Page 11: at the bottom; explain what is intrinsic UV fluorescence.

Response: thanks so much for this comment, and I have added the detail.

Figure 7A: ATP only Control needs to be added to the graph. Figure 7B: ATP only control, and Dye only control are missing in the graphs: please add this data. Can the authors also add data melting curves of FUS RRM with ADP and AMP?

Response: thanks so much for this comment, and we have added the control data in the Figure 7B.

For ADP and AMP, we have just performed the experiments as kindly suggested.

Unfortunately we found that even with ADP and ATP at 1 mM, the RRM samples formed gel-like condensates after the thermal unfolding, thus giving very strange curves in the figure below. One possibility could be the significant reduction of the polarity and solubility of ADP and AMP without one and two phosphates respectively.

Discussion section:

The first half of the discussion repeats the results explained before and is thus redundant. Instead the authors should discuss their findings more in detail in the context of other previous publications looking at the liquid-liquid phase separation and fibrillization of FUS and the role of the different FUS domains in the cell. What are the concentrations in the cell, when does FUS RRM plays a role, how could the detected binding of ATP in neurons be related to pathological FUS in ALS and FTD, etc?

Response: thanks so much for the kind comments. Now I have extensively revised the discussion to incorporate the comments.

Reviewer #3 (Remarks to the Author):

1. Line 230-237: Please show the NMR data that supports the claims in this paragraph.

Response: thanks so much for this comment, and we have added them in Figure 6B.

2. Line 238 and following paragraph: “We also asked the question whether ATP can bind all well-folded proteins.” This question is way too broad sweeping. The profilin-1 NMR data showing that its folded domain does not bind ATP does not add any value to the paper, since it is exemplifying a single point, arguably the exception. I suggest the authors either remove or rethink this section.

Response: thanks so much for this comment, and I have moved them into Supplementary Figure 4, and also revise the description.

3. Lines 257-268: Please include the supporting ThT fluorescence, NMR, microscopy and EM data supporting the claims presented in this paragraph.

Response: thanks so much for this comment, and I have added them as Supplementary Figure 5. We used DIC microscopy to check the samples every day. However, the samples were homogenous solution and thus nothing in the view field. Even so, we checked the samples by EM at both day 5 and day 15 but found no formation of fibrils.

4. Lines 318-320: Please discuss whether any residues from the identified FUS-RRM ATP binding site overlap with the residues within RRM that are associated with fibrillization.

Response: thanks so much for this comment. Unfortunately, as it is extremely challenging to determine the structure of the fibrils formed by FUS RRM, we do not have the knowledge of residues involved in forming fibrils. Previous NMR characterization of fibrillation only showed the intensity reduction and then disappearance of almost all HSQC peaks during the incubation, because upon forming oligomers, NMR peaks became too broad to be detected.

5. Line 332: “[...] RRM domain is absolutely needed to manifest cytotoxicity (11)” This statement is inaccurate. In ref (11), fig. 3A, a FUS truncation containing the PLD and RRM domain (residues 1-373) was not sufficient to induce toxicity.

Response: thanks so much for this comment, and I have revised all related discussion in the manuscript.

6. Regarding the relevance of the ATP-binding to FUS in reducing RRM-mediated fibrillization, please discuss any published evidence that addresses how often FUS exists in cells in an RNA-unbound state. What is the relevance of the mechanism if the protein is always bound to RNA, which competes off the ATP? Discuss how the data in this manuscript fits in the framework presented in Maharana et al, Science (2018), PMID: 29650702, that RNA buffers the phase separation of aggregation-prone proteins?

Response: thanks so much for the kind comments, and I have completely revised the discussion to incorporate the comments.

REVIEWERS' COMMENTS:

Reviewer #2 (Remarks to the Author):

The authors addressed all issues raised and the changes made helped to largely improve the manuscript.

Reviewer #3 (Remarks to the Author):

Kang et al., have addressed the large majority of the concerns I expressed in the first submission. The manuscript could benefit from a thorough proof-reading for typos and grammatical errors, but other than that, I recommend publication of the manuscript.

As all three reviewers had no further scientific comments and only the third reviewer suggested the proof-reading. So I have proof-read and revised the manuscript also in accordance with the format of Communications Biology.